# Two-Photon Polymerized Poly(2-Ethyl-2-Oxazoline) Hydrogel 3D Microstructures with Tunable Mechanical Properties for Tissue Engineering

**DOI:** 10.3390/molecules25215066

**Published:** 2020-10-31

**Authors:** Steffen Czich, Thomas Wloka, Holger Rothe, Jürgen Rost, Felix Penzold, Maximilian Kleinsteuber, Michael Gottschaldt, Ulrich S. Schubert, Klaus Liefeith

**Affiliations:** 1Institute for Bioprocessing and Analytical Measurement Techniques e.V., Rosenhof, 37308 Heilbad Heiligenstadt, Germany; steffen.czich@iba-heiligenstadt.de (S.C.); holger.rothe@iba-heiligenstadt.de (H.R.); juergen.rost@iba-heiligenstadt.de (J.R.); felixpenzold@gmail.com (F.P.); 2Laboratory of Organic and Macromolecular Chemistry (IOMC), Friedrich Schiller University Jena, Humboldtstrasse 10, 07743 Jena, Germany; thomas.wloka@uni-jena.de (T.W.); maximilian.kleinsteuber@uni-jena.de (M.K.); michael.gottschaldt@uni-jena.de (M.G.); ulrich.schubert@uni-jena.de (U.S.S.); 3Jena Center for Soft Matter (JCSM), Friedrich Schiller University Jena, Philosophenweg 7, 07743 Jena, Germany

**Keywords:** hydrogel, poly(2-oxazoline), two-photon polymerization, mechanical properties, tissue engineering, 3D scaffolds

## Abstract

The main task of tissue engineering (TE) is to reproduce, replicate, and mimic all kinds of tissues in the human body. Nowadays, it has been proven useful in TE to mimic the natural extracellular matrix (ECM) by an artificial ECM (scaffold) based on synthetic or natural biomaterials to regenerate the physiological tissue/organ architecture and function. Hydrogels have gained interest in the TE community because of their ability to absorb water similar to physiological tissues, thus mechanically simulating the ECM. In this work, we present a novel hydrogel platform based on poly(2-ethyl-2-oxazoline)s, which can be processed to 3D microstructures via two-photon polymerization (2PP) with tunable mechanical properties using monomers and crosslinker with different degrees of polymerization (DP) for future applications in TE. The ideal parameters (laser power and writing speed) for optimal polymerization via 2PP were obtained using a specially developed evaluation method in which the obtained structures were binarized and compared to the computer-aided design (CAD) model. This evaluation was performed for each composition. We found that it was possible to tune the mechanical properties not only by application of different laser parameters but also by mixing poly(2-ethyl-2-oxazoline)s with different chain lengths and variation of the crosslink density. In addition, the swelling behavior of different fabricated hydrogels were investigated. To gain more insight into the viscoelastic behavior of different fabricated materials, stress relaxation tests via nanoindentation experiments were performed. These new hydrogels can be processed to 3D microstructures with high structural integrity using optimal laser parameter settings, opening a wide range of application properties in TE for this material platform.

## 1. Introduction

Biomaterials are nonliving materials used for applications in contact with living tissue, organisms, or microorganisms [1]. They are routinely used to mimic, repair, or replace biological materials, such as bone, cartilage, or other tissues. The term “biomaterials” itself opens up a wide field of research as well as engineering and is included in various fields, such as tissue engineering, disease modeling, dental and surgical application, implantology/endoprosthetics, drug delivery, and wound dressing only to mention a few [2]. The main classes of biomaterials are ceramics, metals and alloys, natural and synthetic polymers, and composites of polymers with ceramics or metals [3]. It is important to note that biomaterials fulfill some crucial requirements to make them suitable for application in contact with cells and tissues. All biomaterials should be biocompatible, noncytotoxic, and without inflammatory effect when they are intended to be used in biomedicine or regenerative medicine. Depending on their field of use, various requirements are necessary. Functionalization of the material can introduce biological and physicochemical properties, such as biodegradability [4], cell adhesion [5], delivery of signaling molecules [5,6], vascularization [7], and much more. It is also important that the material can be shaped or processed precisely and individually for site-specific therapeutic interventions. This is possible with solid freeform or rapid prototyping techniques, which are used to fabricate scaffolds to reproduce, for example, the architecture of an extracellular matrix (ECM) of tissue [8]. This includes various different techniques, such as stereolithography (SLA), selective laser sintering (SLS), 3D printing, extrusion-based processes, direct laser writing (DLW), two-photon polymerization (2PP), and others [9,10]. The architecture of a scaffold can influence the behavior of surrounding cells. The porosity can allow cells to migrate into the scaffold and initiate the formation of new ECM. In addition to the shape, the surface morphology of the scaffold can also play an important role, for example, in the adhesion and differentiation of cells. Another very important requirement is the mechanical behavior of scaffolds in terms of mechanical stiffness, elasticity, or even viscoelasticity [11]. It has been shown that the rigidity of the matrix influences the differentiation of stem cells. The relatively soft matrices, comparable with brain tissue, induce neurogenic differentiation, whereas more stiffer matrices support myogenic pathway, and the relatively rigid matrices lead to osteogenic differentiation [12]. It is well known that not only do tissue-specific cells differentiate specifically or prefer substrates with stiffer or softer properties but that the mechanical environment can even regulate cell function [13,14]. The process of mechanotransduction leads to a wide range of cellular and supracellular reactions, resulting in specific adhesion, spreading, proliferation, and migration behavior due to variation of matrix rigidity. For example, macrophages grown on substrates with high stiffness yield more proinflammatory mediators [15]. In the field of tumor research, it has been shown that the stiffness of the ECM has a considerable influence on cellular behavior [15]. Recent advances in biomaterial research have attracted great attention in the field of synthetic hydrogels in order to mimic the structure and function found in natural ECM. This not only allows the design and execution of systematic studies in cancer genesis and progression but is also an essential requirement to establish the field of tumor tissue engineering (TTE). While the ECM in an early tumor state can act as a tumor growth inhibitor, it supports the invasion and metastasis of cancer cells in later stages [16]. In addition, tumor cell migration can be directional when the corresponding external cues are provided in the form of a gradient of soluble signals or ECM rigidity (durotaxis/haptotaxis) [17,18,19]. There is also no doubt that the mechanical properties of the surrounding tissue material are vital in bone remodeling and repair as these phenomena are intrinsically connected to the complex force balance between bone matrix and bone cells [20]. Tissue strain and fluid shear stress cause cell deformation, which excites specific signaling pathways [21]. Considering the fact that the mechanical properties of natural as well artificial matrix can be sensed by cells through bidirectional interactions, it is extremely important to establish new bioinspired biomaterial platforms that (i) allow fabrication of precise 3D scaffolds with spatial resolution from macro- to nanoscale, (ii) possess ability to adapt the mechanical properties with regard to tissue type, and (iii) provide maximum flexibility for the introduction of bioactive molecules. 

Within the wide range of biopolymers, hydrogels have shown the highest biomedical potential for application as drug and cell carriers, in 3D cell cultivation, and for biofabrication of tissue matrices [22]. By absorbing high amounts of water, hydrogels are able to mimic the natural tissue. 

Among the numerous synthetic polymer materials, poly(ethylene glycol) (PEG) is commonly used in TE and TTE. Beyond that, PEG is intensively used in daily products, such as nutritional supplements in the food industry as a well as in cosmetics. This extensive use leads to occurrence of anti-PEG antibodies and thus causes patient incompatibilities with PEG-based medicinal products [23,24]. 

Poly(2-oxazoline)s (POx), in comparison to PEG, are in their infancy, although they provide numerous advantages for use as synthetic hydrogel in biomaterial research. Beside the fact that POx show a similar “stealth” effect [25,26] and high biocompatibility [27,28,29,30], POx-based hydrogels are relatively new in the field of tissue engineering. Compared to PEG, POx polymers are simpler to synthesize by cationic ring-opening polymerization (CROP). POx-based polymers have the advantage that they can be designed to individual needs by choosing from a variety of initiators, monomers, and end capping agents [31]. This flexibility enables the synthesis of POx-based hydrogels for cultivation of pancreatic cells [32] and fibroblasts [33] or copolymerization to thermosensitive and amphiphilic polymers [34,35]. Before POx macromolecules can be processed by light-induced 3D fabrication techniques, such as 2PP, they have to be functionalized with polymerizable side or end groups. This was first done by Christova et al. by introducing acrylic groups enabling conventional free radical polymerization [36]. In a previous publication, we have shown that it is possible to process 3D microstructured hydrogel objects by 2PP using diacrylated poly(2-ethyl-2-oxazoline) (PEtOx) [37]. Here, we discuss the results of further research on this topic with a special focus on structural accuracy and integrity. We also present a new hydrogel platform based on newly synthesized PEtOx di- and monoacrylate copolymers with variable mechanical properties and a great application potential in the field of matrix engineering for TE and TTE.

## 2. Results and Discussion

### 2.1. Synthesis and Structural Characterization of Poly(2-Oxazoline) Macromonomers

Recently, the synthesis of PEtOx-based macromonomers with two polymerizable acrylic end groups (PEtOx-DA) using 1,4-dibromo-but-2-ene (DBB) as initiator for cationic ring-opening polymerization was reported [37]. Beyond that, it was demonstrated for the first time that PEtOx-DA can be processed by 2PP to obtain microengineered 3D objects.

One way to adjust the mechanical properties of such 3D objects is to change the length of the polymer chains within the bulk volume. Another option is to vary the number of available polymerizable acrylic end groups. Obviously, PEtOx-DA with two polymerizable end groups acts as a crosslinker, and the corresponding polymerization process results in quite a stiff structure from a mechanical point of view. With this background, it seems to be possible to adjust the mechanical properties of such 3D objects simply by mixing macromonomers with only one polymerizable end group with crosslinker molecules having two polymerizable end groups until a homogenous mixture is achieved. In this manner, microengineered 3D objects can be fabricated with variable viscoelasticity/stiffness. For this purpose, monoacrylated PEtOx polymers are required. Weber et al. synthesized acrylated and methacrylated PEtOx macromonomers and studied their polymerization behavior [38,39]. Based on those studies and the end-capping protocol established in our recent work, we aimed to synthesize monoacrylated PEtOx macromonomers with a targeted degree of polymerization (DP) of 10 (PEtOx-A DP10) and 50 (PEtOx-A DP50) in a larger scale (Scheme 1). For the synthesis of monoacrylated PEtOx macromonomers, the well-known initiator methyl tosylate (MeTos) was used [40,41,42].

The progress of the polymerization was followed via ^1^H nuclear magnetic resonance (NMR) spectroscopy. After quantitative conversion of the monomer, the reaction solution was quenched with acrylic acid and triethylamine. By means of ^1^H NMR spectroscopy, the successful end-capping of the oxazolinium chain by acrylic acid was confirmed. The full spectrum of the purified macromonomer PEtOx-A DP10 and the assigned signals are displayed in Figure 1A. Signal “a” displays the protons deriving from the initiator methyl tosylate between the assigned signals for protons of the polymer backbone (signal “b” and “c”). The same applies to vinyl protons of the acrylate group. By comparing the integrals of the mentioned signals, the degree of functionalization was calculated to be 0.99 for PEtOx-A DP10 as well as for PEtOx-A DP50. The covalent attachment of the acrylic end groups was proven by the appearance of signal “e”, which stands for the methylene protons next to the ester group of the acrylic acid. The ^1^H NMR spectrum of PEtOx-A DP50 can be found in the supporting information (Figure A1 in Appendix A).

The successful covalent attachment of the acrylic acid to the oxazolinium chain was also confirmed by means of matrix-assisted laser desorption ionization with time of flight (MALDI-TOF) mass spectrometry (Figure 1B). The spectrum showed an even distribution of singly charged *m/z* species, which were assigned to the sodium-ionized macromolecular species. 

These ionized species revealed regular-spaced *m/z* intervals of 99 g mol^−1^, which corresponds to one monomer repeating unit of EtOx. The inlay showed a calculated species overlayed with a measured one, proving the successful attachment of the acrylate groups to the oxazolinium chain.

For the synthesis of PEtOx-a DP50, all CROP conditions were kept the same as in the synthesis of PEtOx-A DP10. Figure 1C displays the size exclusion chromatography (SEC) overlay of PEtOx-A DP10 and PEtOx-A DP50. The PEtOx-A DP50 species (black curve) showed a lower elution volume compared to PEtOx-A DP10, thus having a higher molar mass and proving the higher DP. The results from MALDI-TOF measurements of PEtOx-A DP50 showed a broad *m*/*z* distribution in the spectrum, which was identified as the sodium-ionized species (see Figure A2b in Appendix A). All data for both macromonomers are summarized in Table 1. 

### 2.2. Swelling Behavior, Mechanical Properties, and 2PP Structuring of Different POx Systems

Our previous processing results showed that it is possible to process POx by 2PP, and the voxel and line dimensions can be changed by a corresponding variation of the laser parameters. Nevertheless, POx is a hydrogel and naturally underlies swelling effects under aqueous conditions. This basic driving force of hydrophilic polymers or hydrogels to absorb large quantities of water without dissolving results from an osmotic force induced by hydrophilic functional groups covalently coupled to the polymer backbone. This osmotic pressure is balanced by the crosslink density, which limits the stretching of the polymer network and prevents its deformation. Its strength is determined by the rubberlike elasticity of Gaussian chains [43]. The Gibbs free enthalpy of elastic deformation, $∆G_{el}$, scales with the deformation ratio as follows:(1)$$∆G_{el}\;\approx \;N_{s}^{-1}\propto^{2}$$
where $N_{s}^{-1}$ is the number of segments between two successive crosslinks, and ∝ is the linear deformation ratio. At the equilibrium, both forces (osmotic forces and elastic restoring forces) are balanced [44].

In this study, the influence of the initial POx precursor composition using PEtOx mixtures of different DP and number of polymerizable groups on the final mechanical stability was thoroughly evaluated. Additionally, the swelling behavior of copolymers with different chemical compositions was considered. Therefore, mixtures based on two different PEtOx-DAs, which are already described elsewhere [37], and newly synthesized PEtOx-A as shown and discussed before, were prepared in order to gain maximum variability with regard to the mechanical properties of the final 3D objects produced. The prepared POx mixtures are listed in Table 2. All formulations, including PEtOx-DA with DP10 are designated as POxA, while those with DP20 are designated as POxB. A commercially available diacrylated poly(ethylene glycol) (PEG-DA) with a molar mass of 700 g/mol corresponding to a DP of about 13 was chosen for comparison mainly to the PEtOx-DA with a DP of 10.

For the selected POx mixtures, always one diacrylated and one monoacrylated PEtOx were chosen and mixed with a ratio of 80:20 and 20:80 weight percent to obtain maximum difference with regard to the macromonomer lengths on the one hand and to ensure a sufficient gap with regard to chemical structure and the associated mechanical stability between the pure PEtOx-DA or PEtOx-A systems and the new mixtures on the other hand. It is of value to point out that the final precursor systems for the photopolymerization reactions were prepared by adding the same amount of ultrapure water as the weight of the macromonomer to get a processable (2PP-compatible) viscous liquid. With respect to the macromonomer weight, 0.5 wt% of photoinitiator IRGACURE^®^ 2959 for UV curing and 0.5 wt% of BA740 for 2PP structuring were added. The chemical structure and the crosslinking efficiency of the photoinitiator BA740 is described in detail elsewhere [37,45].

Cylindrical samples of 6 mm diameter and 4 mm height were prepared by UV curing of the different precursor systems for investigation of the degree of swelling as well as the mechanical properties of the compositions. 

In Figure 2, the results of the swelling investigations are shown in two diagrams displaying the swelling of the samples caused by the amount of absorbed water (Figure 2a) and the associated change of the sample dimensions after storage in ultrapure water (Figure 2b,c). 

As expected, all UV-cured material systems absorbed more or less water. The grey dashed line marks the amount of initial water content in the precursor, which should still be present in the sample after the curing process. The amount of additionally absorbed water varied between about 3% up to 37% depending on the actual copolymer of the selected precursor system. The lowest value of about 3% was found for POxA1, which was the pure diacrylated PEtOx with a DP of 10. This can be explained by a higher amount of chain links per volume unit, which usually results in a higher density of the polymer network and therefore decreased swelling in water. With this background, it seems to be understandable that the swelling degree increased with the addition of monoacrylated PEtOx (POxA2 and POxA3) due to a decrease in the crosslink density per unit volume. Accordingly, the highest amount of absorbed water was found for POxA4, which contained the longest monoacrylated PEtOx. A further increase up to 80 wt% of monoacrylated species (POxA5) destabilized the polymer structure, so partial dissolution phenomenon was observed. This can be explained by the low number of chain links and therefore the reduced crosslink density in combination with a very high macromolecular chain length. This allowed much more water molecules to diffuse into the bulk structure, and after a certain point, the structural integrity of the sample was lost. Basically, the swelling behavior of the copolymer systems based on PEtOx-DA with a DP of 20 showed similar tendencies with only few differences in comparison to the copolymers with a DP of 10. The pure diacrylated system POxB1 already showed an additional absorbed amount of water of about 10% after swelling, which was due to the lower chain link density compared to POxA1. According to our expectations, the amount of water then further increased by addition of short-chain monoacrylated macromonomer (POxB2 and POxB3). When the monoacrylated macromonomer with DP50 was added (POxB4), the additional amount of water was about 14%. A further increase in the concentration of monoacrylated macromonomer with DP50 up to 80 wt% (POxB5) led to nearly complete loss of structural integrity after swelling due to the reduced crosslink density. It should be mentioned that, from a point of view of swelling theory, it is highly unlikely that a small change of crosslink density of monoacrylated PEtOx on the one hand and a small change of the chain length of monoacrylated PEtOx on the other hand will have the same impact on the observed swelling behavior. It is worth noting that the swelling behavior observed for the different copolymers depends strongly on the crosslink density and the macromolecular chain length as can be expected from the theoretical considerations based on the Gibbs free enthalpy of elastic deformation, $∆G_{el}$, as a first approach. 

The reference sample based on PEG-DA revealed an uptake of about 15% additional water after UV curing, which is slightly more than the sample made of POx with a similar degree of polymerization (POxA1). The impact of swelling on the sample dimensions, namely, diameter and height, showed a similar behavior and therefore led to the same conclusions.

At this point, it seems to be reasonable to assume that (i) the observed swelling behavior and (ii) its dependence on the chemical composition and crosslink density must have a tremendous effect on the structural integrity/accuracy of a 3D object produced as well as on its stability and mechanical properties. 

For mechanical evaluation, the samples were measured by stress relaxation via nanoindentation, which has become an important tool for measurement of mechanical properties at small scales to gain more insight on the viscoelastic behavior of different materials. The obtained relaxation functions were phenomenological by nature, as can be seen in Figure 3a with PEtOxA2 as an example. The curves were then fitted with a continuous viscoelastic model comprising a sum of two logarithmic normal distributions of relaxation times *H* (fitting can be seen in Figure 3b) following the equations below:(2)$$H\;\left[\log\left(\tau \right)\right]=H_{1}\;\left[\log\left(\tau \right)\right]+H_{2}\;\left[\log\left(\tau \right)\right]\;\mathrm{where}\;H_{1,2}\;\left[\log\left(\tau \right)\right]=C_{1,2}\frac{b_{1,2}}{\sqrt{\pi }}\;exp^{\left(-b_{1,2}log\frac{t}{\tau_{1,2}}\right)}$$
with $\tau_{1,2}$ as the (unknown) relaxation time and the parameters $C_{1,2},\;b_{1,2}$ as additional unknowns in the fitting procedure. The free parameters that need to be determined during the fitting procedure followed the assumption that the investigated hydrogels exhibited a very broad relaxation spectrum; their detailed molecular analysis was not the subject of the present study.

In addition to a parallel spring for $E_{0}$, seven unknowns were selected to be calculated via a genetic optimization algorithm: minimum and maximum of the storage modulus, maximum of the loss modulus and its frequency and value of the storage modulus at the point of maximum of the loss modulus, and height and half width of dispersion. A simplified scheme of the relaxation model is shown in Figure 3e. The results are summarized in Table 3 for all material systems.

The methodical approach for fitting the data obtained from stress relaxation measurements is a simulation-based computational optimization technique, which implements a nondeterministic search method that emerges from evolutionary algorithms (EAs). Relevant examples are evolutionary strategies, evolutionary programming, and genetic algorithms (GAs) [46,47]. These techniques can iterate several solutions simultaneously and combine the most promising ones to generate a new, improved set of solutions. EAs have been successfully applied to solve problems with nonlinear or even discontinuous objective functions, nonconvex objective function spaces, as well as to poorly conditioned problems [48,49,50]. The computational techniques take advantage of iterative applications of random variations and subsequent customized selection over a population of prospective solution instances. A GA does not require detailed knowledge of the problem; it only requires a dedicated method to compare different solutions based on the respective fitness function in order to select the best solutions and continue the evolutionary process until a certain convergence criterion is met. For detailed information on this topic, we refer to the relevant literature [51,52].

The fit of the relaxation function allows calculation of the relaxation time spectrum, as can be seen in Figure 3c. With the relaxation time spectrum, other material functions, such as storage modulus $\left(E^{\prime }\right)$ and loss modulus $\left(E^{\prime \prime }\right)$ at various frequencies, can be calculated via Equations (3) and (4). Figure 3d shows the dynamic moduli calculated by means of the continuous viscoelastic model based on the sum of two logarithmic normal distributions of relaxation times.
(3)$$E^{\prime }\left(\omega \right)=E_{0}+\overset{\infty }{\underset{-\infty }{\int }}H\;\left[log\left(\tau \right)\right]\;\left(\frac{\omega^{2}\tau^{2}}{1+\omega^{2}\tau^{2}}\right)\;d\left(\log\left(\tau \right)\right)$$
(4)$$E^{\prime \prime }\left(\omega \right)=\overset{\infty }{\underset{-\infty }{\int }}H\;\left[log\left(\tau \right)\right]\;\left(\frac{\omega \tau }{1+\omega^{2}\tau^{2}}\right)\;d\left(\log\left(\tau \right)\right)$$
where ω is the angular frequency. In this way, it was possible to allocate a specific constitutive equation to each copolymer system for comparison purposes. It should be noted that all parameters exhibited a material-specific frequency with respect to time dependence.

In order to be able to (i) identify the most important factors of influence that govern the highly variable viscoelastic behavior and (ii) gain a frequency-independent parameter, principal component analysis (PCA) was performed [53]. In our case, PCA showed that 83% of the data variance was included in the first two principal components (PC1 = 65% and PC2 = 18%), while all other PCs could be neglected (see plot in Table 3). The PCs corresponded to the directions with the maximum amount of variation in the data set. It is important to note that PCA caused a significant data reduction and allowed a clear statement about the extent to which the investigated copolymers followed a viscoelastic approach defined with Equations (3) and (4). As an example, a boxplot of the first PC considering the complete set of the investigated copolymer systems is shown in Figure 4 together with some key parameters of the chosen viscoelastic approach.

According to the data in Table 3, the first PC obviously displayed the same tendencies compared to the key parameters of the selected viscoelastic constitutive equation, e.g., maximum of the storage modulus, minimum of the storage modulus, or storage modulus at maximum of the loss modulus. As mentioned above, the established hydrogel platform was based on two different PEtOx-DAs and the newly synthesized monoacrylated poly(2-ethyl-2-oxazoline)s (PEtOx-A) designated as POxA and POxB, respectively.

For example, the influence of the amount of monoacrylated PEtOx was clearly detectable, resulting in lower values of the first PC for POxA2 and POxA3 compared to POxA1 and for POxB2 and POxB3 compared to POxB1. In the same manner, the longer monoacrylated PEtOx showed a lower mechanical strength than the corresponding copolymers with the shorter monoacrylated PEtOx (POxA4 vs. POxA2 and POxB4 vs. POxB2). 

All these lead to the following conclusions: (i) the established hydrogel platform based on poly(2-oxazoline)s (POx) possess the potential to design new biopolymers with highly defined and variable mechanical properties by varying the crosslink density and the macromolecular chain length and (ii) the chosen continuous viscoelastic approach based on the sum of two logarithmic normal distributions of relaxation times *H* is excellently suited to describe the mechanical behavior of the newly synthesized hydrogels. 

After identifying the superior performance of the established hydrogel platform by means of UV-cured samples (single-photon absorption), it was necessary to show that POx can also be processed to 3D microstructures by 2PP (two-photon absorption) [37]. As usual, a material-specific parameter search protocol had to be elaborated in terms of structural integrity and accuracy, taking into consideration especially the swelling behavior of hydrogels. For this purpose, the POx systems POxA1-5 and POxB1-5, as well as the PEG-DA for comparison purposes, were processed to so-called “spider web arrays”. A 3D computer-generated model of the spider web structure is shown in Figure 5. Its base was a hexagonal ground plate, and the surface was patterned by bars of different thicknesses (1, 2, 4, 6, and 8 µm) arranged in a star-shaped spider web manner.

To identify the optimal parameter setup for the 2PP process, the arrays were processed with varying writing speeds in the X-direction from 1 to 7 mm/s in steps of 2 mm/s and with varying laser power in the Y-direction of 200 to 400 mW in 100 mW steps. Four arrays were processed for every copolymer, differing in the line distances in the X- and Y-directions of 1, 2, 4, and 6 µm, while the line distance in the Z-direction was kept constant at 5 µm. Figure 6 shows confocal laser scanning microscopy (CLSM) images of the four arrays processed with POxA2 with different line distances. Corresponding figures of all other material systems can be found in the supporting information (Figure A1, Figure A2 and Figure A3 in Appendix A). 

Remarkably, it was found that it was possible to produce 3D structures by means of all the applied parameter sets and with different line distances. With decreasing laser power and increasing writing speed, the structures lost their structural integrity, which was also the case for high laser power in combination with low writing speed. This can be explained by the energy required to initiate the polymerization process. If the laser energy remains below a certain threshold, the photoinitiator cannot be transformed into the excited state. Following that, no radicals can be formed, and therefore no polymerization takes place. To overcome this threshold, the laser power must be increased, while the writing speed should be decreased to introduce sufficient energy per time into the focal volume. Otherwise, when the energy input is too high, the polymerization process is no longer limited to the focal volume and involves a larger surrounding volume, which results in overpolymerization and again in a loss of structural integrity and accuracy. 

Regarding the structures in Figure 6 in terms of different line distances, a line distance above 1 µm resulted in structures where single lines could be observed, which means that no densely crosslinked 3D object was formed. In the current case, this made it impossible to investigate the swelling behavior and the identification of the optimal parameter set. Therefore, only spider web arrays with a line distance of 1 µm in the X- and Y-directions were generated for further investigation.

For the evaluation of structural accuracy and integrity, new arrays of 16 spider web structures with varying laser power from 100 to 400 mW at 100 mW steps and varying writing speed from 1 to 7 mm/s in 2 mm/s steps with line distances of 1 µm in the X- and Y-directions and 5 µm in Z-direction were prepared. After the array production, CLSM images of all arrays were taken. In order to find the parameter set with the best lateral contour conformity, it was necessary to determine structural accuracy and integrity by comparing every single processed structure and the dimensions of the original CAD model. The established evaluation process comprised four main steps: (a) postprocessing of the CLSM image, (b) binarization, (c) comparison with the CAD model, and (d) a scoring step to identify the best 2PP setup. In Figure 7, this procedure is documented for PEtOxA2 as an example. The corresponding data of all other copolymers can be found in the supporting information (Figure A6, Figure A7 and Figure A8 in Appendix A).

In the postprocessing step (a), the CLSM images were vertically and horizontally aligned and color scaled corresponding to the grey scale of the confocal image. The images were then binarized and lateral rescaled to have the same X–Y dimensions as the source CAD model (b). This step was necessary due to different swelling ratios of the polymers depending on the precursor used. Subsequently, the images were subtracted from the source CAD model (c), and the underdimensioned and overdimensioned pixels were summed-up for scoring (d). The scoring finally led to the optimal parameter set for the respective precursor in terms of maximum contour conformity. Following this strategy, each score value was composed of two factors: under- and overdimensioned pixels. For this purpose, the cumulative percentage of over- and underdimensioned pixels were each divided into eight classes and assigned a specific value, as shown in Table 4.

Because an incompletely polymerized structure must be an exclusion criterion for the respective parameter set, the percentage of underdimensioned pixels was used as the most important quality criterion. As a second, comparatively weak criterion, the percentage of overdimensioned pixels was used due to the fact that small overdimensioned regions are acceptable as long as all structural features are represented with sufficient quality. However, this criterion can achieve a higher impact on the score value at deviations larger than 60%, with overdimensioned regions leading to an unacceptable geometrical failure.

For classification, these two values were added together. In this way, an individual score was obtained for each precursor and each parameter set, with the respective minimum value representing the optimal parameter set. By implementing the method mentioned above for all precursors, a specific optimized parameter setup could be identified. 

The results are shown in Table 5.

A general correlation between the energy input and the structure conformity was poor. However, it is noticeable that the POxA1 and POxB1 precursor systems had different thresholds for high structural conformity. For all material systems based on PEtOx-DA DP10, a poor structural conformity was found for 100 mW power, whereas some based on PEtOx-DA DP20 showed the best performance at 100 mW. Similarly, the highest and lowest writing speeds were not applicable to most material systems and led to either significant over- or underdimensioning. Overall, the parameter sets for best performance were in the range of 33 to 133 J/mm energy input, with higher energy inputs causing suboptimal results for all the investigated formulations.

## 3. Materials and Methods 

### 3.1. Chemicals and Instrumentation

Triethylamine (TEA, Sigma Aldrich, St. Louis, MO, USA) was distilled under argon atmosphere prior to usage. Acrylic acid was purchased from Sigma Aldrich (Darmstadt, Germany) and was used without further purification. 2-Ethyl-2-oxazoline (EtOx, Sigma Aldrich) was distilled to dryness over calcium hydride (VWR, Radnor, PA, USA) under argon atmosphere prior to usage. MeTos (Sigma Aldrich) was distilled to dryness over barium oxide (BaO, Acros, Fair Lawn, NJ, USA) and stirred under argon atmosphere prior to usage. Acetonitrile was obtained from a solvent purification system (MB-SPS-800 by MBraun) and stored under argon. ^1^H and ^13^C NMR spectra were measured on an AC 300 (300 and 75 MHz; Bruker, USA) spectrometer at 298 K. Chemical shifts are reported in parts per million (ppm, *δ* scale) using the signal of the deuterated solvent as reference. For measurements of MALDI-TOF mass spectra, an Ultraflex III ToF/ToF instrument (Bruker Daltonics, Bremen, Germany) was used. The instrument is equipped with a Nd: YAG laser and a collision cell. All spectra were measured in the positive reflector mode using *trans*-2-[3-(4-*tert*-butylphenyl)-2-methyl-2-propenylidene] (DCTB) or dithranol as matrix and sodium iodide (NaI) as doping salt. The instrument was calibrated prior to each measurement with an external poly(methyl methacrylate) (PMMA) standard (2500 g mol^−1^) from Polymer Standard Services (PSS). Size exclusion chromatography was measured on an Agilent 1200 series system equipped with a PSS degasser, a G1310A pump, a G1362A refractive index detector, and a PSS GRAM guard/30/10 Å column series running with *N*,*N*-dimethylacetamide (DMAc) with 0.21% lithium chloride. The Techlab oven was set to 50 °C, and the molar masses were calculated using a PMMA standard (505 to 981,000 g mol^−1^) from PSS.

### 3.2. UV Curing and 2PP Structuring of 3D Scaffolds

Eleven different precursor mixtures were prepared as samples for mechanical analysis as well as for the 2PP structuring, namely, five different formulations each based on the two different PEtOx-DAs (DP of 10 or 20) and one PEG-DA for reference purposes (M_n_ = 700 g mol^−1^, Sigma Aldrich). Four out of five formulations based on the PEtOx-DA precursors were mixtures with PEtOx-As (DP10 or DP50) in weight percent ratios of 80:20 and 20:80 for each PEtOx-A macromonomer. The fifth formulation represented the pure DA macromonomer. For example, the precursor consisting of PEtOx-DA DP10 and MA DP10 in a weight percent ratio of 80:20 was prepared by 1.6 g (1.35 mmol) of PEtOx-DA DP10 and 0.4 g of PEtOx-A DP10 (0.37 mmol). In the case of PEtOx-A DP10, the macromonomer was dried in vacuo before usage because of its high hygroscopy. All other PEtOx mixtures were prepared accordingly. Then, 2 g ultrapure water was added to obtain processable solutions. Even if the PEG-DA was already in a liquid processable form, water was added to get a proper comparison. Finally, 10 mg of the photoinitiator IRGACURE^®^ 2959 (0.004 mmol, Sigma Aldrich) for UV curing or BA740 (0.02 mmol) for 2PP structuring were added to the mixtures. All mixtures were stirred overnight at room temperature to ensure complete dissolution. 

The UV curing was performed with a Vacuum UV Exposure Unit 2 from proMa systro, which works at a wavelength of 365 nm and power of 120 W. Silicone masks for cylindrical samples with height of 4 mm and diameter of 6 mm were placed in the UV chamber, filled with the precursor, and covered with glass slides to get a flat sample surface. Subsequently, curing was performed for 30 min.

The 2PP system M3DL from Laser NanoFab GmbH (Hannover, Germany) was equipped with (i) a Ti:sapphire femtosecond laser working at 800 nm wavelength, delivering 140 fs pulses at 80 MHz repetition rate (VISION II, Coherent, Scotland); (ii) a beam splitter combination consisting of a cube polarizer and a λ/2-plate for attenuation of the laser power in the 2PP experiment; (iii) a CCD (charge-coupled device) camera for monitoring the process; and (iv) an acousto-optic modulator for rapid on/off switching of irradiation. The beam was focused by an ×50 objective lens with NA 0.75 (EC EPIPLAN, Zeiss, Germany). Before the 2PP experiments were performed, the precursor/PI mixtures were placed in a home-made 1 mm thick silicone frame on a glass slide equipped with 3-(trimethoxysilyl) propyl methacrylated surface functionalization (Sigma Aldrich, Germany) to improve adhesive strength between the glass slide and the 2PP structure. A second glass slide on top formed the upper cover plate. The 3D structure was produced by moving the focal volume of the laser beam with an ultraprecise linear motor-driven three-axis positioning stage according to the computer model. The stage movement was coupled to a Galvano scanner (ALB 10100 Air Bearing Stage, Aerotech Inc., Pittsburgh, PA, USA) to increase the writing speed and shorten the necessary processing time. Arrays of spider web structures were processed with increasing laser power from 100 to 400 mW in 100 mW steps in one dimension (X) and increasing writing speed from 1 to 7 mm/s in 2 mm/s steps in another dimension (Y). Furthermore, arrays differing in line distances in the X- and Y-directions of 1, 2, 4, and 6 µm were produced, while the line distance in the Z-direction was kept constant at 5 µm. For imaging purposes, ground plates of 1.5 × 1.5 × 0.1 mm (line distances of 10 µm in the X- and Y-directions and 5 µm in the Z-direction) were processed for arrays with line distances of 1 µm before the process started because the arrays delaminate from the glass slide when the material is soaked in water during development. To remove the nonilluminated precursor after irradiation, the samples were eluted in ultrapure water for 24 h as developer (acetone was used for PEG-DA). The developer was replaced periodically according to common laboratory practice.

### 3.3. Synthesis of PEtOx-A by CROP

Polymerizations yielding PEtOx-A in different lengths (DP10 and DP50) were carried out in a 1 L Normag reactor equipped with a KPG (Kerngezogenes Präzisions-Glasgerät) stirrer. The reactor was washed with 800 mL of boiling acetone for 1 h, dried at 115 °C under an argon stream for 1 h, evacuated, and filled with nitrogen thrice. 

#### 3.3.1. General Synthesis Procedure

MeTos, EtOx, and acetonitrile were pumped into the reactor in that order. Afterward, the solution was refluxed, and the end of the polymerization was determined by ^1^H NMR spectroscopy by taking 1 mL aliquots in regular intervals. After cooling to 40 °C, acrylic acid was first added, followed by triethylamine, and the mixture was stirred overnight at 50 °C. The reaction mixture was allowed to cool to room temperature, and a sample was analyzed by means of ^1^H NMR spectroscopy. The crude product was then poured into a 2 L round bottom flask, leftover residue in the reactor were rinsed with 250 mL acetonitrile, and the solvent was removed under reduced pressure. After that, the crude product was dissolved in 1 L dichloromethane, washed thrice with a saturated NaHCO_3_-solution and thrice with brine, and dried over Na_2_SO_4_/MgSO_4_ (30/1). Subsequently, the product was filtered, and the solvent was removed under reduced pressure. The final product was dried in vacuo for an additional day to remove all the leftover solvent. 

#### 3.3.2. Synthesis of PEtOx-A DP10

Weighing: MeTos (30.4 mL, 202 mmol), EtOx (204 mL, 2.02 mol), acetonitrile (270 mL), acrylic acid (20.7 mL, 303 mmol), and triethylamine (56 mL, 404 mmol). Yield: approximately 190 g (the final product was stored in CHCl_2_ because of its hygroscopic properties).

^1^H NMR (CDCl_3_, 300 MHz): *δ*6.29 (br dd, *J* = 17.14, 3.20 Hz, 1H), 6.11–5.88 (m, 1H), 5.87–5.69 (m, 1H), 4.17 (s, 2H), 3.34 (br s, 37H), 2.91 (s, 3H), 2.38–2.11 (m, 20H), and 1.00 (br s, 30H). SEC (DMAc, 0.21% LiCl, RI detection, PS calibration): M_n_ = 2030 g mol^−1^, Ð: 1.18. 

PEtOx-A DP50: Weighing: MeTos (6.1 mL, 40.3 mmol), EtOx (204 mL, 2.02 mol), acetonitrile (294 mL), acrylic acid (4.15 mL, 60.5 mmol), and triethylamine (11.2 mL, 80.7 mmol). Yield: 184 g.

^1^H NMR (CDCl_3_, 300 MHz): *δ*6.33−6.20 (m, 1H), 6.04−5.91 (m, 1H), 5.82−5.68 (m, 1H), 4.15 (br s, 2H), 3.41–3.27 (br s, 200H), 2.94–2.87 (br s, 3H), 2.33−2.11 (m, 115H), and 0.98 (br s, 149H). SEC (DMAc, 0.21% LiCl, RI detection, and PS calibration): M_n_ = 7150 g mol^−1^, Ð: 1.16.

### 3.4. Synthesis of PEtOx-DA by CROP and Photoinitiator BA740

The crosslinker macromonomer PEtOx-DA was synthesized according to reference [37] and the photoinitiator BA740 according to reference [45].

### 3.5. Mechanical Measurements, Imaging, and Evaluation

Nanoindentation experiments were carried out with a MFP-3D nanoindenter from Asylum Research. The cylindrical samples were submersed in water to prevent drying effects and indented by a spherical sapphire tip with a diameter of 500 µm. All indentation experiments were performed with an initial indentation path of 10 µm, which was reached after 0.01 s. Then, the indentation was stopped, and the stress relaxation was monitored for 20 s with a sampling rate of 1 kHz. The static Young´s modulus for each timepoint was calculated implementing the Hertz model.

CLSM pictures were taken with a LSM 710 confocal laser scanning microscope from ZEISS in ultrapure water using a tile scan mode to achieve larger pictures of higher quality. Pictures were taken at one focal plane; sometimes, the sample had to be lifted slightly on one side to obtain a proper image of all existing structures processed by 2PP. The necessary fluorescence signal for the CLSM imaging originates from the photoinitiator attached covalently to the polymer system, which makes staining unnecessary. Therefore, a higher intensity means a higher density of polymerized volume. 

Statistical calculations and figures for data evaluation of the mechanical measurements were performed using the programming language “R” [54]. A program implementing the outlined procedure was written in Fortran and CUDA C/C++ language. The algorithm is controlled by a set of operational parameters. A series of preliminary empirical essays was carried out to determine a suitable set of parameters, which caused the algorithm to converge in a reasonable time.

The postprocessing and analysis of the spider web arrays was done with the software SPIP™ (Image Metrology, Lyngby, Denmark), while a software plugin developed by Francesco Marinello (DIMEG Lab, University of Padova, Padova, Italy) was used for binarization.

## 4. Conclusions

In this study, successful preparation of hydrogels based on poly(2-ethyl-2-oxazoline)s with different mechanical properties was carried out via two-photon polymerization. Different formulations based on newly synthesized PEtOx macromonomers of different chain lengths and crosslink densities combined with suitable photoinitiators served as precursor materials. Samples processed via UV curing were characterized regarding their degree of swelling as well as their viscoelastic behavior via nanoindentation. For this purpose, a continuous viscoelastic model based on the sum of two logarithmic normal distributions of relaxation times was adapted and validated by multivariate data analysis for the first time to describe the mechanical behavior of PEtOx hydrogels. 

Depending on the formulation, samples with a higher crosslink density per volume in general revealed a lower tendency for swelling and higher stiffness. The use of macromolecules with greater chain lengths as well as a higher volume fraction of macromolecules with reduced crosslink density caused a higher degree of swelling and reduced stiffness of the hydrogels until a loss of structural integrity was observed. 

In addition, different material systems were processed by two-photon polymerization to 3D microstructures, which showed excellent structural integrity. The resulting structures were investigated by lateral contour conformity evaluation based on confocal laser scanning microscopy. By means of this new evaluation strategy, material-specific parameter sets could be identified to guarantee proper structural conformity between 2PP-fabricated structures and the CAD model.

It is worth emphasizing that hydrogels based on poly(2-ethyl-2-oxazoline)s constitute a promising approach in the field of matrix engineering due to their structural and mechanical properties. It is assumed that scaffolds made of these polymers will mimic the natural ECM of numerous tissues, especially as poly(2-ethyl-2-oxazoline)s can be structured precisely by 2PP with excellent structural integrity and tailor-made stiffness. This favors these hydrogels for hard and soft tissue applications with a special focus in the field of tumor tissue engineering.

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
