# Peer review of "Two-Photon Polymerized Poly(2-Ethyl-2-Oxazoline) Hydrogel 3D Microstructures with Tunable Mechanical Properties for Tissue Engineering"

_molecules, 2020, doi:10.3390/molecules25215066_

Round 1

Reviewer 1 Report

COMMENTS [Article ID: molecules-973011]
This paper reports a study concerning hydrogels produced via two-photon polymerization. This study encompasses a characterization of the properties of the produced hydrogels concerning the swelling behaviour, mechanical properties and structural integrity. It is recommended for publication in Molecules after minor revision indicated below.

  1. Revise the overall manuscript since there are some typos.
  2. The main novelty of the study should be mentioned clearly.
  3. Please describe better the mechanical evaluation concerning nanoindentation (indentation time, material for the indentor, analytical measure used, etc.).
  4. Please unify the format of the references. For example, page numbers from reference number 9 was included as “745-59”, whereas for other references appears complete.

Reviewer 2 Report

The work by Czich and colleagues introduce a hydrogel system based on poly(2-ethyl-2-oxazoline)s that can be polymerized via 2-photon radiation.  They utilize an interesting evaluation approach to assess structural fidelity of polymerized constructs in a binarized manner. While the work has several significant and innovative aspects, there are a number of minor issues that the authors should address in order to further improve the quality of the data presentation.

1- Figures 2-3: are presented too large, with font sizes that are too large and not proportional to the rest of the manuscript text size. Also, I suggest combining figures 1-3 into one figure with multiple panels to consolidate all characterization data in one place.

2- Please report 'n' for all the graphs and qualitative data presented in the manuscript. Further, statistical analysis must be performed for all quantitative assays and reported in the figures and figure captions. For instance, in figure 4 plots, stats are missing. Same story in the Figure 6.

3- Make sure to properly introduce all the acronyms that are used in this study, which is quite a few. For instance, '2PP', 'ECM', and 'CAD" in the Abstract have not been properly defined.
